# Psychological Impact of *TP53*-Variant-Carrier Newborns and Counselling on Mothers: A Pediatric Surveillance Cohort

**DOI:** 10.3390/cancers14122945

**Published:** 2022-06-15

**Authors:** Amanda Scartezini Gozdziejewski, Clarice Wichinescki Zotti, Isabela Aparecida Moreira de Carvalho, Thairine Camargo dos Santos, Luana Rayana de Santi Walter, Karin Rosa Persegona Ogradowski, Karin Luiza Dammski, Heloisa Komechen, Monalisa Castilho Mendes, Emanuelle Nunes de Souza, Mariana Martins Paraizo, Ivy Zortea da Silva Parise, Guilherme Augusto Parise, André Luiz Grion, Gislaine Custódio, Rosiane Guetter Mello, Bonald C. Figueiredo

**Affiliations:** 1Pelé Pequeno Príncipe Research Institute, 1532 Silva Jardim, AV., Curitiba 80250-200, PR, Brazil; scartezini.amanda@gmail.com (A.S.G.); clawzotti@gmail.com (C.W.Z.); karin.persegona@fpp.edu.br (K.R.P.O.); kdammski@gmail.com (K.L.D.); heloisakomechen@gmail.com (H.K.); monalisacmendes@gmail.com (M.C.M.); emanuelle.nuness@gmail.com (E.N.S.); mmparaizo@gmail.com (M.M.P.); ivyparise@gmail.com (I.Z.S.P.); gaparise@gmail.com (G.A.P.); algrion@yahoo.com.br (A.L.G.); 2Faculdades Pequeno Príncipe, 333 Iguaçu Av., Rebouças, Curitiba 80230-902, PR, Brazil; morebela.isabela@gmail.com (I.A.M.C.); camargo.thairine@gmail.com (T.C.S.); luana.santi@ufpr.br (L.R.S.W.); 3Centro de Genética Molecular e Pesquisa do Câncer em Crianças (CEGEMPAC), UFPR, 400 Agostinho Leão Jr. Av., Curitiba 80030-110, PR, Brazil; custodio.gislaine@gmail.com

**Keywords:** *TP53* p.R337H mutation, predictive testing, genetic counselling, anxiety, depression

## Abstract

**Simple Summary:**

Children who inherit a *TP53* (tumor suppressor) mutation tend to develop adrenocortical tumor (ACT) in the initial years of life, requiring genetic counseling and DNA testing after birth. In 2001, a *TP53* mutation (R337H) was identified in Paraná State (South Brazil) and was later confirmed to be highly frequent in South Brazil, where the pediatric ACT incidence is up to 20 times higher than that in other countries. We evaluated anxiety and depression in mothers of these newborns to determine the effect of genetic testing. We found that anxiety (but not depression) is presented by mothers of R337H-positive newborns for less than 4 months. These findings were more common in mothers with previous mental health vulnerabilities. The anxiety test scores must be differentiated, and appropriate psychological support should be provided for these mothers and for mothers of newborns inheriting other types of *TP53* mutation in other countries.

**Abstract:**

Counselling and genetic testing (CGT) after neonatal screening may increase depression and anxiety (DA) levels during cancer surveillance. This study assessed the DA scores in mothers of newborns from Paraná state, Southern Brazil, carrying the *TP53* p.R337H variant. To understand and adjust DA conditions during term of pregnancy, we initially detected sociodemographic covariates [marital status (MS), number of children (NC), and/or education level (EL): MS-NC-EL] on an independent group of pregnant women (not subjected to genetic testing). The Hospital Anxiety and Depression Scale (HADS) was used to assess risk factors in pregnant (cross-sectional analysis) and unrelated mothers (at 2-month intervals, longitudinal study) of *TP53* p.R337H-tested newborns (three sessions of HADS analysis) using Wilcoxon (Mann–Whitney) and Kruskal–Wallis nonparametric tests. Lower anxiety levels were observed in mothers of noncarriers (without MS-NC-EL = 6.91 ± 1.19; with MS-NC-EL = 6.82 ± 0.93) than in mothers of p.R337H carriers in the first session (without MS-NC-EL = 6.82 = 8.49 ± 0.6025, with MS-NC-EL = 6.82 = 9.21 ± 0.66). The anxiety levels significantly decreased 4 months after CGT (third session) in mothers of p.R337H carriers. We did not find a significant change in depression scores. Mothers with mental health instability requiring medications need periodical psychological support during and after CGT.

## 1. Introduction

Individuals harboring the germline *TP53* p.R337H founder variant display increased risk for various cancers characterized by marked clinical heterogeneity. Carriers exhibit a distinctive pattern of early-age onset of adrenocortical tumors (ACT) at an occurrence rate of 80–95% of all pediatric ACTs in South and Southeast Brazil [1,2]. They also exhibit an increased risk of developing pediatric choroid plexus carcinoma, neuroblastoma, sarcoma, and other cancer types in adults in non-consistent proportions [3,4,5,6].

Classical Li-Fraumeni syndrome (LFS) is an inherited familial cancer predisposition syndrome including soft tissue sarcoma or osteosarcoma in childhood or young adulthood [7]. Li-Fraumeni-like syndrome (LFL) shares many of the LFS features, but the age and type of specific cancers seen in affected family members follow a less strict criteria [8,9]. Furthermore, LFS criteria have also become more flexible as named “modified Chompret criteria” [10]. Despite ACT being diagnosed in only <5.0% of all p.R337H-carrier newborns before 15 years of age [2,11], and being more frequent in other *TP53* variants with complete penetrance (i.e., classical LFS) [12], all these families seem to require psychological support before, during, and after genetic counseling.

*TP53* p.R337H mutation is a founder effect [13] clustered in South and Southeast Brazil, while only a few carriers have been reported in other countries (reviewed by [14] Pinto et al., 2020). It is less pathogenic than *TP53* variants in the DNA-binding domain (DBD) of the protein, which are most probably associated with the classical LFS [7]. Individuals harboring DBD mutations, independent of the protection at social or environmental levels where they live, are prone to an earlier onset of cancer and a higher cancer incidence than those less pathogenic germline variants [15,16,17]. The reason behind finding the absence or presence of a family history of cancer among p.R337H families is not clear, showing that clinically validated classification is still uncertain for this variant [13]. New insights have emerged from the discovery of p.R337H co-segregation with another tumor suppressor mutation, *XAF1* p.E134* in 17p13.1, in approximately 70% of the newborns in Brazil [18]. This second variant may increase the cancer risk of sarcomas; however, it is probably much less pathogenic for the onset of ACT [18]. Moreover, the exposure to environmental pollutants as a risk of additional somatic variants should be taken into consideration [19].

Most p.R337H carriers are inaccurately estimated for their cancer risk; therefore, genetic counselling and education should be personalized for identifying the pattern of genotype and cancer segregation to determine the extent of cancer risk in each family (reviewed in the Appendix A). The impact of this information, as well as genetic testing, may cause anxiety or depression prior to psychological analysis of the mothers; therefore, necessary psychological support during and after genetic counseling should be provided. To the best of our knowledge, no study has focused on the impact of germline mutation testing in newborns on mothers’ behavior; therefore, our understanding is limited to studies involving adult participants in genetic testing for cancer risk.

In other countries, different *TP53* variants predominate, usually causing classic LFS with higher frequency of ACT, brain tumors, and sarcomas at pediatric age [12,16,17]. *TP53* p.R337H is hypomorphic (i.e., a low-penetrance variant in more than 50% of our large cohort) [2,5,14] in contrast to the lifetime cancer risk of germline *T**P53* gene variants in classic LFS carriers, affecting over 70% of males and nearly 100% of females [20]. This difference is important to minimize emotional impact during counseling, particularly for those families inappropriately labeled as classic Li-Fraumeni. Most patients, of all sociodemographic backgrounds, independently try to Google and understand the impact of being p.R337H carriers, and usually find the difference between simple “familial cancer” (i.e., not LFS) and LFS. Therefore, certainty about their mutation status and cancer risk is necessary with full-pedigree data analysis in genetic counseling. Notably, a surveillance protocol using periodical exams for *TP53* carriers would not be cost effective for all p.R337H carriers at all ages due to time-consuming follow-ups, but was proved useful for preclinical and clinical diagnosis of ACT in the first 5 years after birth using a simplified surveillance protocol [11]. It is also during the first 5 years of age that other reported cancers may occur among p.R337H carriers, such as plexus choroid carcinoma and neuroblastoma [3,4].

Counselling and genetic testing (CGT) are performed to identify germline mutations associated with predispositions for certain types of cancers in families with probands under follow-up. However, there is a great deal of debate regarding levels of depression and anxiety (DA) brought about by CGT. General distress (from the State Anxiety Scale) and test-specific distress (a 15-item scale that measures responses to genetic test results) were used to evaluate short-term psychological responses to testing of *BRCA1* mutation (associated with a high risk of breast and ovarian cancers) in participants from a Mormon kindred [21]; a significant increase in test-specific distress in carriers compared to that in noncarriers were observed, even when in-person genetic counseling was provided. Furthermore, the authors claimed that adverse psychological effects of genetic testing might predominate in individuals with less health knowledge and risk awareness [21]. Other factors possibly involved in genetic counselling, such as pretest stress variables, expectations [22], environment and social support network [23], sociodemographic variables [24], family and personal history of cancer [25], and the psychologist’s presence during counselling for emotional support [26], must be evaluated. Most authors believe in the beneficial effects of CGT, such as in predicting the development of a probable disease and, consequently, providing preventive procedures against advanced cancer [23]. These authors are primarily concerned with advice given to families with high cancer risk, the recommended programs of surveillance, and primary prevention. Comparing the results may be difficult for different tests. For example, a meta-analysis based only on tests for *BRCA1/2* variants identified anxiety and cancer-specific distress among the carrier participants [24]; soon after receiving test results, carriers’ emotional distress increased slightly and returned to the pretesting levels with additional time, whereas the noncarriers experienced decreases in both general and cancer-specific distress soon after testing.

Despite the diversity of predictive tests, results, and participants among different studies, the extent to which DA affects mothers of newborns carrying a germline variant is not yet known. The Guthrie test (i.e., newborn blood spot screening test) for the *TP53* p.R337H variant was performed for cost-effective, low-intensity surveillance without periodical medical and laboratory exams [2,11]. The first p.R337H test in a newborn is performed after precounselling, together with all Guthrie tests, 2–3 days after birth. Our goal was to assess DA levels in the mother immediately after counselling, disclosure of the first test, and blood draw to confirm the germline *TP53* p.R337H variant in her newborn using the polymerase chain reaction–restriction fragment-length polymorphism (PCR-RFLP) method, together with confirmation of the mutation in newborn and parents using the DNA sequencing method [2]. The interval between the two DNA methods was 1–2 weeks. In addition, after CGT and before DA assessment, the parents receive instructions about the signs and symptoms of ACT and other types of pediatric cancer in three sessions. We performed a longitudinal study to evaluate DA levels at three different time points at 2-month intervals. As a secondary goal, we also determined differences in anxiety and depression scores related to cancer cases in carriers or death due to cancer in the family in the last 12 months, and differences when mothers were not p.R337H carriers (i.e., in the absence of the potential guilt complex). The relationship between sociodemographic indicators and high psychological impact, particularly close to the term of pregnancy, is controversial. We hypothesize that any of the covariates (marital status, number of children, employment, education level, and family income) evaluated in a neutral group of pregnant women (i.e., without interference of genetic testing for cancer) may increase DA. If true, the covariate should be adjusted for the groups of mothers of newborns tested to determine the cancer risk.

## 2. Materials and Methods

### 2.1. Design and Participants

Prior to establishing criteria for evaluating the psychological impact of CGT in mothers of tested newborns, we performed a cross-sectional analysis on an independent group of pregnant women (not subjected to genetic testing) unrelated to the mothers of the screened newborns) between the 30th and the 34th week of gestation to estimate the typical perinatal DA status. The rationale for including HADS-A and HADS-D assessments in pregnancy is that they are associated with the changes in hormonal and psychological functions, such as mood changes, which may range from anxiety to depressive reactions. If our hypothesis is true, the covariate associated with significant changes in DA (marital status, number of children, employment, education level, or family income) will be adjusted accordingly to the group of mothers of the tested newborns. The study was approved by two ethics committees, one from Faculdades Pequeno Príncipe (CAAE number 49409715.7.0000.5580) and the other from Secretaria Municipal de Saúde de Curitiba (CAAE 49409715.7.3001.0101), and all participants signed the informed consent. The study was conducted in the State of Paraná (Brazil) in 2017 and 2018.

To assess depression and anxiety in pregnant women, a cross-sectional survey was developed using the Hospital Anxiety and Depression Scale (HADS), validated in Portuguese [27]. The sample comprised 734 pregnant women who were between 30 and 34 weeks of gestation and who were taking prenatal care, aged 18 or over, and who did not use medication for anxiety, depression, or toxoplasmosis, and who were not at risk of premature birth.

### 2.2. Recruitment of p.R337H-Carrier Newborns through Neonatal Screening

A neonatal screening was designed to re-evaluate the *TP53* p.R337H gene frequency, regional variation in adrenocortical tumor rates, and establish a surveillance protocol without periodical exams to confirm adrenocortical tumor cure with diagnosis and surgery at early-onset tumor stage I [11]. All participants provided written informed consent. The study was approved by the Ethics Committee of Pequeno Príncipe Hospital and National Research Ethics Committee (CAAE number 50622315.0.0000.0097).

### 2.3. Longitudinal Analysis—Assessment of the Psychological Impact of the TP53 p.R337H Mutation Carrier Status of Newborns on Mothers

Perinatal DA risk factors identified in unrelated pregnant women (marital status, number of children and education level) were used to adjust the control and experimental groups of mothers. Precounselling was performed within 2–3 days of birth by a trained nurse at the hospital, immediately before the Guthrie test (*TP53* p.R337H). Parents were invited by phone for CGT and confirmatory tests (7–14 days after the p.R337H *TP53* test) (Figure 1). After counselling (see Appendix A) and psychological support, the mothers were invited for the anxiety and depression assessments. Mothers of carrier (experimental group) and noncarrier (control group) newborns were assessed using the Hospital Anxiety and Depression Scale (HADS) score for anxiety and depression levels. These two groups were further subdivided to separate mothers with any of the sociodemographic covariates based on the analysis on an independent group of pregnant women not exposed to DNA tests or genetic inquiries. The full method, including genetic counselling, is described in the Appendix A. The study design included physicians, nurses, and psychologists (Figure 1). The experimental E1 and control C1 groups of mothers were classified based on the absence of any covariate (marital status, education level, or number of children), while the presence of any covariate was considered as E2 or C2.

The validated Portuguese version of the HADS [27] was used to assess the levels of anxiety (HADS-A) and depression (HADS-D) at each day of visit. Each scale is composed of seven items, and the highest score is 21, corresponding to greater anxiety and depression. In the present study, we followed the recommended cut-off values: normal (0–7), borderline (8–10), and disturbance (11–21). In addition, a sociodemographic form evaluated age, place of residence, occupation, civil status, level of education, number of children, family income, and family history of cancer and other diseases, including psychiatric or psychological treatment. Cases presenting with high scores (≥11) and/or other signs and symptoms of mental disorders were scheduled for further consultations with psychologists. Control mothers (C1 and C2) who received a negative result for *TP53* p.R337H were recruited and assessed by telephone.

### 2.4. Data Analysis

Wilcoxon (Mann–Whitney) and Kruskal–Wallis nonparametric tests and generalized linear models with Box–Cox power exponential distribution were used to identify the conditions associated with anxiety and/or depression levels in the cross-sectional analysis of data from pregnant women. The identified conditions associated with high HADS scores in pregnant women were used to compare medians of anxiety and depression levels in two or more groups, respectively [28]. The Box–Cox power exponential distribution has four parameters, and positive real number support [29]. This distribution was chosen because it maximized the likelihood of models of anxiety and depression levels without explanatory variables. The significance of the model components was obtained by the likelihood ratio test.

Generalized estimating equations with an unstructured covariance matrix were used to model the population-averaged anxiety and depression levels of mothers in a longitudinal study. The response variables were modelled as a function of time, group, and the interaction between them. The estimated marginal means (and standard errors) were obtained from these models. The significance of the model components was tested using Wald’s statistics. Multiple comparisons among means were made using pairwise contrasts, and *p*-values were corrected using Bonferroni’s method. The software environment for statistical computing and graphics R version 3.3.3, 2007 [30] was used for the data analyses with the gamlss package for the generalized linear models [31] and geepack package for generalized estimating equations [30].

## 3. Results

### 3.1. Cross-Sectional Analysis in Women Close to Term of Pregnancy—Assessment of the Sociodemographic Covariates, Such as Marital Status, Number of Children, Employment, Education Level, and Family Income

All mothers of tested newborns lived in Paraná (South of Brazil), and no statistical difference was found in their mean ages of 26.73 ± 3.96 and 26.12 ± 6.57 (mean ± standard deviation, SD), for the control groups (non-p.R337H carriers) C1 (without any sociodemographic variable associated with anxiety or depression) and C2 (with any sociodemographic variable associated with anxiety and/or depression), respectively; no statistical difference was found in mothers’ mean ages of 25.99 ± 6.07 and 27.91 ± 5.28 (mean ± SD) for the experimental groups (p.R337H carriers) E1 (without any sociodemographic variable associated with anxiety or depression) and E2 (with a sociodemographic variable associated with anxiety or depression), respectively.

This is a distinct group of women (unrelated to the evaluation of genetic testing), which is important for a neutral analysis because it presents common characteristics (effects associated with pregnancy, the sociodemographic and ethnical background) with our main experimental group presented in Section 3.2. The HADS scores between the different characteristics are presented in Table 1. The factors that presented significant *p*-values (*p* < 0.05) for anxiety and depression levels were number of children (*p* < 0.001), marital status (*p* < 0.001), and education level (*p* < 0.003). These analyses led to the identification of characteristics that increase HADS score at the end of pregnancy: the number of children (≥2 for HADS-A—Appendix A; and ≥1 for HADS-D—Appendix A), unmarried (for HADS-A and HADS-D—Appendix A, respectively), and elementary school (for HADS-A and HADS-D—Appendix A, respectively).

### 3.2. Longitudinal Analysis—Assessment of the Psychological Impact of p.R337H Mutation Carrier Status of Newborns on Mothers

A total of 77 of the invited mothers of the newborns positive for p.R337H (77/117, 66%) and only 22 (22/120, 18%) of the noncarriers’ mothers (control) agreed to participate. Mothers in the control (C1 = 11 and C2 = 11) and experimental (E1 = 35 and E2 = 42) groups completed the three HADS sessions (T1, T2, and T3) included in the analyses. The participant characteristics are listed in Table 2. Disclosure of the genetic test took place approximately 10 days after the first HADS assessment. All mothers of tested newborns lived in Paraná, and no statistical difference was found in their mean ages of 26.42 and 27.03 years for the control (non-p.R337H) and experimental (p.R337H carriers) groups, respectively.

We found no significant difference in anxiety (mean HADS-A) between Day 1 (T1, session 1) and Day 120 (T3, session 3) in the control groups (C1/T1 vs. C1/T3, *p* = 1.000; C2/T1 vs. C2/T3, *p* = 1.000), suggesting that anxiety does not occur in the absence of p.R337H carrier newborns. We found a significant difference in anxiety (mean HADS-A) between Day 1 (T1) and Day 120 (T3) in the experimental groups (E1/T1 vs. E1/T3, *p* < 0.0001; E2/T1 vs. E2/T3, *p* < 0.0113), suggesting that anxiety is of short duration based on the result disclosure of a positive test for *TP53* p.R337H (Figure 2).

We found no significant difference in depression (mean HADS-D) between Day 1 (T1) and Day 120 (T3) of the control and experimental groups (Figure 3). We found no difference between control and experimental groups: mean HADS-A: T1/C1 (6.91 ± 1.19) vs. T1/E1 (8.49 ± 0.6025), *p* = 1.000; T1/C2 (C2 = 6.82 ± 0.93) vs. T1/E2 (9.21 ± 0.66), *p* = 0.2109; T3/C1 (6.36 ± 1.03) vs. T3/E1 (6.02 ± 0.57), *p* = 1.000; and T3/C2 (6.27 ± 0.93) vs. T3E2 (7.38 ± 0.60), *p* = 1.000. The decrease in anxiety scores from T1 to T3 was greater in E1 (2.46 ± 0.54) and E2 (1.83 ± 0.63).

Mean HADS-D scores were ≤7 in all groups and timelines, that is, below the pathological cut-off (Figure 3). HADS-A and HADS-D scores among mothers who transmitted p.R337H to their newborns (N = 34) were not significantly different from noncarrier mothers (N = 43). Similarly, there was no impact of recent cancer cases in carriers or death due to cancer (<12 months) in the family on HADS-A and HADS-D scores.

## 4. Discussion

This is the first longitudinal study applied to mothers in response to a test of positive cancer predisposition for a germline *TP53* (tumor suppressor) variant in their newborns using HADS-A and HADS-D. Approximately 50% of the newborns inherit this variant from their mothers. During counselling, parents learn that the probability of developing ACT is variable depending on the penetrance of each variant, with complete penetrance causing classical LFS [12] or low penetrance such as *TP53* p.R337H presenting lower number or no cancer cases in more than 50% families [2]. Associated genetic variants inherited from each parental side may exist (further details in the Genetic Counselling Protocol in the Appendix A). HADS tests in mothers focused on negative (C1 and C2) and positive p.R337H tests (E1 and E2) in newborns. The duration of the higher detected level of anxiety was less than 4 months after the first assessment (T1, Figure 1). The early impact (only anxiety) detected in the present study is in line with the acute short-duration outcome results reported in a systematic review, wherein the predictive test (PT) and the psychological test were performed in genetically affected adults [22]. We did not find any difference in HADS-A and HADS-D scores between p.R337H-carrier (N = 34) and noncarrier mothers (N = 43) four months after test disclosure.

Bosch et al. (2012) [32] did not observe significant anxiety or depression levels at 3 months and 1 year after the disclosure of the BRCA1/2 test among those identified as carriers. These authors claimed that the allegedly negative impact of PT does not occur in most tested individuals; however, they have not considered the possibility of an earlier impact. In a study by Kasparian et al. (2009) [33], carriers of the CDKN2A mutation displayed a significant decrease in HADS-A scores 2 weeks after disclosing the results, suggesting that in some cases, anxiety is acute and of short duration. We decided to assess anxiety and depression immediately after genetic counselling, after blood drop sampling from the finger (parents) and the heel (newborns) to confirm positive genetic testing in newborn, and after instructions about signs and symptoms of pediatric cancer. Thus, our third evaluation (4 months later—T3) did not detect significant differences between controls (mothers of noncarriers) and mothers of p.R337H-carrier newborns. Similarly, using the State–Trait Anxiety Inventory (STAI) instrument for Lynch syndrome, anxiety level was high 1 week after the test result, but did not differ from baseline levels between carriers and noncarriers at 1 month, 1 year, and 7 years later [34]. Other authors have also used STAI and reported that anxiety levels did not differ between carriers and noncarriers at 3 years after PT for hereditary nonpolyposis colorectal cancer (HNPCC) [35]. In another study, distress levels in patients tested positive for HNPCC increased slightly in carriers after disclosure and decreased 6 months later [36].

Distress levels did not differ between carriers and noncarriers 3 and 6 months [37], 6 and 12 months [38], and 3 years after performing PT for BRCA1/2 [37]. Thus, despite the differences in instruments or the measured score, recovery to baseline values takes place before 4 months. The recovery timing has been systematically reviewed; for example, as per a meta-analysis, BRCA1/2 PT distress increased immediately after receiving the test result and decreased 5–24 weeks later [24]. It was suggested in a systematic review that the test is rarely a predictor of distress for more than a month [39]. In contrast, a Canadian study assessed Jewish women submitted to BRCA1/2 and found that distress increased one year after receiving the genetic test result [40]. According to these authors, distress declined between 1 and 2 years in women who underwent prophylactic surgery.

The measured scores in genetically affected individuals may be influenced by the baseline values of HADS-A and HADS-D scores [32,41] and distress [37,38]. Several factors may influence baseline values, and we have demonstrated that marital status, education level, and number of children may influence anxiety in pregnant women (not subjected to genetic testing). Higher income was associated with less distress [37] but family income did not change the outcome in our study design. This was partly attributed to the small variation in the economic condition among the families, and/or the cultural/ethnical background of the population. Despite a lack of difference in mean HADS-A levels between the control and experimental groups 4 months after disclosure, the baseline level remained high (score > 10) in C1 (18.2%), C2 (11.4%), E1 (11.4%), and E2 (19%). Similar data were reported in 14% of BRCA-tested women [32]. In another study, a higher percentage of participants (31% of noncarriers and 26% of carriers) presented with abnormal STAI levels two years after testing for Lynch syndrome [42], which concurs with another study [36]. One year after the PT for BRCA 1/2, the authors detected that 16.7% of the participants presented with severe levels of anxiety [43].

A significant proportion of our participants reported psychological symptoms before CGT due to family and personal problems (34% in the experimental group), and some participants reported the use of psychiatric medications (7.7%). One or more of these problems were observed in mothers with high HADS-A scores (>11). Taken together with the detected risk factors identified in pregnant women (marital status, number of children, and education level), it is important to filter these possible interferences in CGT.

The levels of depression assessed in our study (T1, T2, and T3) were classified as not clinically relevant (<8). In addition, no significant difference in HADS-D was found between mothers of carrier and noncarrier newborns. This result corroborates the study showing low levels of HADS-D from the pretest to 6 months [44]. In another longitudinal study, HADS-D levels did not differ between carriers and noncarriers 3 years after disclosure of HNPCC [35]. In addition, a PT study for the CDKN2A/p16 mutation in minors also found no relevant clinical levels for depression using STAI children [45]. The low percentage of patients presenting with depression (score > 8) found in the present study in groups E1 (2.9%), E2 (16.7%), C1 (9.1%), and C2 (9.1%) corroborate a study showing low levels of depression (2%) observed three months after PT for BRCA [32]. However, contrary to our findings, in a study carried out for Lynch syndrome, the authors found that 25% of participants presented with moderate depressive symptoms 2 years or more after the result disclosure [42].

In our study, mothers who had a cancer case in a carrier or had lost a relative due to cancer in the last 12 months did not present with a significant difference in HADS-A or HADS-D in relation to mothers without cancer reported in the same group. Other studies have reported different results. For example, carriers that did not have a personal history of cancer had higher distress scores six months after the result, compared with noncarriers with and without a history of cancer [46]. Conversely, higher levels of distress were found in people with a greater number of first-degree relatives diagnosed with cancer for PT for Li-Fraumeni syndrome [47].

We acknowledge the limitations of our current study, which include the lack of previous research studies (i.e., emotional impact to the mother for risk of cancer or other diseases in the newborn); the lack of similar analysis in carrier fathers (less available) and first- or second-degree relatives; the lack of assessment of the psychological wellbeing with other tests (e.g., Impact of Event Scale—IES) and coping strategies; and the lack of a large sample size for better statistical measurements. However, these limitations have not affected our results, conclusions, or interpretations. Despite these limitations, it is necessary to consider comments about risk factors in this study and to be cautious about CGT after neonatal screening leading to a positive result disclosure. The means of support for the mothers of p.R337H carrier newborns need a longitudinal design, as the one used, to establish the necessary temporal sequence for the study. Therefore, in future investigations, the completion of such a longitudinal study could be simplified to focus only on HADS-A (only two tests, at day 1 and day 120) and to examine the possible role of interventions focused on detecting previous psychological conditions that may further aggravate emotional reactions during and after CGT. These recommendations would establish the effective strategies in CGT methods more precisely. Given the lack of similar studies, the aforementioned findings should be explored further. HADS-A or other similar instruments are valuable in quantitatively evaluating the differences between control and experimental participants. Our contribution is to help psychologists to understand the evolution of symptoms over time and the potential harm of an anxiety of short duration. The HADS-A and HADS-D findings in pregnant women offer significant potential for research. Parents of any background are capable of perceiving the psychologist’s support and how this positively influences their perception is tailored to individual needs and their carrier relatives. In addition, future studies could evaluate symptomatology of post-traumatic stress disorders in parents facing a disclosure of a genetic test positive for detrimental diseases in their newborns. For example, IES was performed in men and women after predictive test for *BRCA 1/2* [38]. Similarly, we believe that future studies are needed to address resilience and coping strategies.

## 5. Conclusions

To the best of our knowledge, this is the first analysis on the psychological impact for mothers after disclosing a genetic test positive for cancer predisposition in their newborns. Anxiety lasts less than four months and is predominantly mild, except in mothers with previous mental instability. Our second main finding was the three sociodemographic variables associated with DA (marital status, number of children, and education level), highlighting the importance of considering these variables with high susceptibility to anxiety, which may interfere with the psychological analysis of other predictive tests. With respect to application in practice, psychologists could use only the HADS-A test at two time points (i.e., on day 1 and day 120), while considering the possibility of other vulnerabilities associated with pre-existing emotional instability or mental disorders.

## Figures and Tables

**Figure 1 cancers-14-02945-f001:**
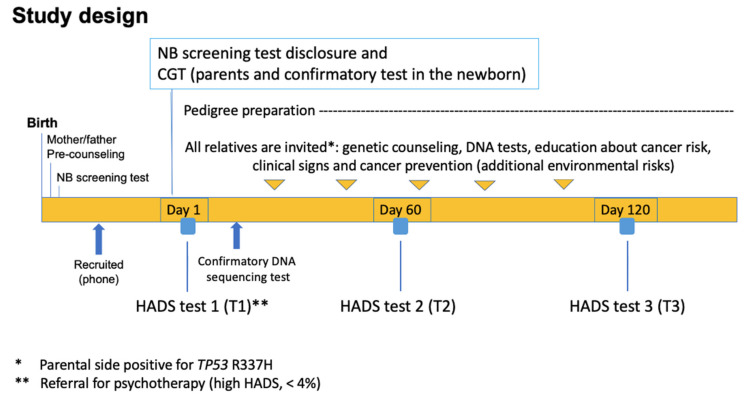
Study design flowchart. After precounselling and the genetic test on newborns at the hospital by trained nurses, parents of experimental groups were invited (1–2 weeks after birth) to obtain test results and receive counselling, wherein a second DNA test was conducted on their newborns to confirm *TP53* p.R337H. After result disclosure of the newborn’s first test, finger (parents) and heel prick (newborn) blood drop sampling, counselling, and education of parents about signs and symptoms of pediatric cancer, the parents were invited for three HADS tests at 2-month intervals. The timeline of clinical and psychological protocols was monitored by telephone. Control group mothers (C1 and C2) who received a negative result for the *TP53* p.R337H in their newborns were recruited and assessed by telephone.

**Figure 2 cancers-14-02945-f002:**
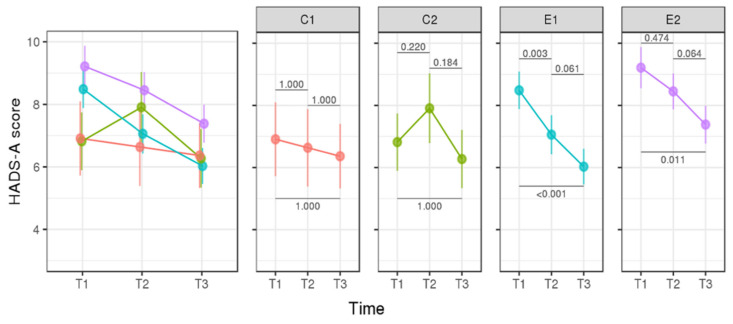
HADS-A (anxiety) score in mothers of noncarrier (control group) and carrier newborns (experimental group). Detected risk factors in pregnant women were considered (bias groups, C2, and E2). *p*-values with Bonferroni correction are shown between the groups (horizontal lines).

**Figure 3 cancers-14-02945-f003:**
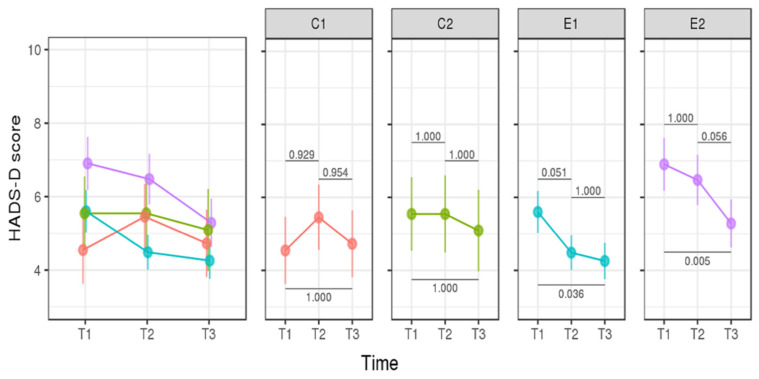
HADS-D (depression) score in mothers of noncarrier (control group) and carrier newborns (experimental group). Detected risk factors in pregnant women were considered (groups with any sociodemographic covariate, C2 and E2). *p*-values with Bonferroni correction are shown between the groups (horizontal lines).

**Table 1 cancers-14-02945-t001:** Anxiety and depression HADS scores according to the levels of each characteristic among pregnant women.

Characteristics	Level	HADS-A Level	A *p*-Value	HADS-D Level	D *p*-Value
**Number of children (N) ^1^**			** *p* ** **< 0.001 ^2^**		** *p* ** **< 0.001 ^2^**
	0	5.00 (0.19)		3.56 (0.17)	
	1	5.45 (0.14)		4.14 (0.13)	
	2	5.89 (0.20)		4.72 (0.19)	
	3	6.33 (0.31)		5.30 (0.304)	
	4	6.78 (0.44)		5.88 (0.43)	
	5	7.22 (0.58)		6.46 (0.56)	
** Employment **			* p * = 0.716		* p * = 0.503
	No	6 [3, 9]		4 [2, 7]	
	Yes	6 [3, 9]		4 [2, 7]	
** Marital status **					** * p * ** ** < 0.001 **
	Married	5 [3, 9]		4 [2, 6]	
	Not married	7 [4, 1]		6 [3, 9]	
** Education level **			** * p * ** ** < 0.006 **		** * p * ** ** < 0.003 **
	Elementary school	7 [3, 1]		5 [3, 9.8]	
	High school	5 [3, 9]		4 [2, 7]	
	Undergraduate	5 [2, 8.5]		3 [1.5, 8]	
	Graduated	5 [3, 7]		3.5 [2, 6]	
** Family income monthly **			* p * = 0.249		* p * = 0.254
	A (EUR < 221.79) *	6 [3, 11]		5 [3, 8]	
	B (EUR 221.79–443.58) *	6 [3, 10]		5 [2, 7]	
	C (EUR 443.58–665.37) *	5 [3, 8]		3.5 [2, 6]	
	D (EUR > 665.37) *	5 [4, 7]		4 [3, 6]	

1. For the number of children, measures are the estimated medians (and standard error) obtained by the generalized linear model. For other characteristics, the measures are the observed medians (and interquartile range). 2. *p*-value of the likelihood ratio test for number of children. For other characteristics, *p*-values are from nonparametric tests as Wilcoxon (Mann–Whitney), when there are two levels (employment and marital status), and Kruskal–Wallis, when there are more than two levels (education level and family income). * (EUR 1 = BRL 4.81 at the time of data collection) HADS: Hospital Anxiety and Depression Scale.

**Table 2 cancers-14-02945-t002:** Characteristics identified in mothers of tested newborns.

	Level	Control N (%)	Experimental N (%)	* p *
N		22	77	
Mean age (years ± SD)		26.42 ± 5.30	27.03 ± 5.70	>0.65
Number of children	0	8 (36.4)	33 (42.9)	0.497
	1	11 (50.0)	27 (35.1)	
	2	3 (13.6)	7 (9.1)	
	3	0 (0.0)	6 (7.8)	
	4	0 (0.0)	2 (2.6)	
	5	0 (0.0)	2 (2.6)	
Employment	No	14 (63.6)	45 (58.4)	0.848
	Yes	8 (36.4)	32 (41.6)	
Marital status	Married	20 (90.9)	62 (80.5)	0.413
	Not married	2 (9.1)	15 (19.5)	
Education level	Elementary school	8 (36.4)	29 (37.7)	0.851
	High school	10 (45.5)	31 (40.3)	
	Undergraduate	2 (9.1)	5 (6.5)	
	Graduated	2 (9.1)	12 (15.6)	
Family income (monthly)	A (EUR < 221.79)	4 (18.2)	17 (22.1)	0.577
	B (EUR 221.79–443.58)	11 (50.0)	27 (35.1)	
	C (EUR 443.58–665.37)	4 (18.2)	23 (29.9)	
	D (EUR > 665.37)	3 (13.6)	10 (13.0)	

## Data Availability

The data that support the findings of this study are available from the corresponding author upon reasonable request.

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
