# Peer review of "Psychological Impact of TP53-Variant-Carrier Newborns and Counselling on Mothers: A Pediatric Surveillance Cohort"

_cancers, 2022, doi:10.3390/cancers14122945_

Round 1

Reviewer 1 Report

Abstract  

  • Line 33-37: The type of study design needs to be clearly mentioned (cross-sectional or longitudinal).
  • Need to state the place/country where the study conducted, and the time of sampling took place.
  • Need to state the type of statistical analysis used.

 Introduction  

  • Line 50-52; Line 71-73: What about other countries? Is there any evidence?
  • Line 77-86: Description of studies is vague, suggest to better define the study type, so that the reader can evaluate quality.
  • Line 90-97: These systematic reviews/meta-analysis contain little synthesis of the information. Please provide more information about the results of these studies. For example, what are the type of tests used? (Line 92-93).
  • Line 115-118: This is unclear. Authors should provide an in-depth explanation for why it needs to be examined-why do they think it matters? It is preferable to make the hypothesis strongly grounded.

 Method  

  • Please mention the time of sampling took place.
  • Line 122-125: This should be clearly described.
  • Line 139-155: This should be moved to results.
  • Line 194: Please clarify if the Portuguese version used is valid or reliable.
  • Please move Table 2 to results.
  • Regarding to longitudinal data analysis, it is unclear what was the drop-out rate? How authors deal with missing values? 

 References  

  • Some references are very old (Ref #7,8,14,17,20,22,23,35,36)- please update.

Author Response

Reply

We have modified the introduction part to provide sufficient background of the study undertaken in the manuscript.

The revised version has been edited by Editage for language, consistency, and flow of information.

Comments and Suggestions for Authors

Abstract  

  • Line 33-37: The type of study design needs to be clearly mentioned (cross-sectional or longitudinal).
  • Need to state the place/country where the study conducted, and the time of sampling took place.
  • Need to state the type of statistical analysis used.

Reply
We have added the type of study design (cross-sectional and longitudinal study), the location in Brazil, and the statistical methods used.

 Introduction  

  • Line 50-52; Line 71-73: What about other countries? Is there any evidence?

Reply

We have included relevant data about the TP53 variants in other countries (lines 65-67 and 89-92) and improved the background regarding the hypomorphic TP53 p.R337H to explain the importance of genetic counseling to differentiate less than 50% of families with the Li-Fraumeni syndrome (potentially more associated with emotional impact) in lines 67-102.

  • Line 77-86: Description of studies is vague, suggest to better define the study type, so that the reader can evaluate quality.

Reply

This section has been improved with additional information (now lines 110-121, as well as 67-102).

  • Line 90-97: These systematic reviews/meta-analysis contain little synthesis of the information. Please provide more information about the results of these studies. For example, what are the type of tests used? (Line 92-93).

Reply

We simplified and provided the relevant information about references 23 and 24 (lines 121-130).

  • Line 115-118: This is unclearAuthors should provide an in-depth explanation for why it needs to be examined-why do they think it matters? It is preferable to make the hypothesis strongly grounded.

Reply:

We agree, we have replaced it (lines 148-154)

“The relationship between sociodemographic indicators and high psychological impact, particularly close to the term of pregnancy, is controversial. We hypothesize that any of the covariates (marital status, number of children, employment, education level, and family income) evaluated in a neutral group of pregnant women (i.e., without interference of genetic testing for cancer) may increase DA. If true, the covariate should be adjusted for the groups of mothers of newborns tested to determine the cancer risk.”

 Method  

  • Please mention the time of sampling took place.

Reply: The study was conducted in 2017 and 2018, in the State of Paraná (Brazil) (lines 172-173).

  • Line 122-125: This should be clearly described.

Reply: The last sentences in the introduction (lines 154-158) and in methods (lines 158-169) were modified to explain why we conducted the analysis on pregnant women.

  • Line 139-155: This should be moved to results.

Reply

The data has been moved to Results.

  • Line 194: Please clarify if the Portuguese version used is valid or reliable.

Reply

The Portuguese version of the HADS has been validated as reported [now reference 27]. It is valid and reliable.

  • Please move Table 2 to results.

Reply

The data has been moved to Results.

  • Regarding to longitudinal data analysis, it is unclear what was the drop-out rate? How authors deal with missing values? 

Reply

Detailed information has been added (lines 284-287).

“Seventy-seven of the invited mothers of the newborns positive for p.R337H (77/117,  66%) and only 22 (22/120, 18%) of the non-carriers’ mothers (control) agreed to participate. Mothers in the control (C1 = 11 and C2 = 11) and experimental (E1 = 35 and E2 = 42) groups completed the three HADS sessions (T1, T2, and T3) included in the analyses.”

 References  

  • Some references are very old (Ref #7,8,14,17,20,22,23,35,36)- please update.

Reply

We have removed 14 references (including some of the old ones, and other references about genetic testing unrelated to cancer) and introduced 8 new references.

Reviewer 2 Report

This manuscript aims to assess Depression and Anxiety scores in mothers of new-33 borns carrying the TP53 p.R337H variant and to identify the possible socio-demographic associated factors. This paper is interesting, with a particular scientific soundness, well structured and clear. I have only some minor concerns.

Have you analyzed the possible impact of mothers’ age on depression or anxiety? The literature gives some important indications on the possible higher indexes in younger mothers than older ones and the possibility to answer to this research question could be assessed also in this study.

Another important fact that it could be inserted in the discussion could be to assess the Post-Traumatic Stress Disorder symptomatology in these mothers, a possible psychopathology in mothers subjected to such a diagnosis of their children.

A limit is recognizable in the adopted methodology: probably a self-report measure could be subjected to social desirability and other more specific measures to assess psychological wellbeing should be adopted in the future studies.

What about the possible resilience factors in this sense? Depression didn’t emerge as a significant variable. Perhaps resilience or coping strategies could impact on this result.

Author Response

Reply

The manuscript has been modified (highlighted in red). The conclusions section has been rewritten (lines 441-451):

“To the best of our knowledge, this is the first analysis on the psychological impact of mothers after disclosing a genetic test positive for cancer predisposition in their newborns. Anxiety lasts less than four months and is predominantly mild, except in mothers with previous mental instability. Our second main finding was the three sociodemographic variables associated with DA (marital status, number of children, and education level), highlighting the importance of considering these variables with high susceptibility to anxiety, which may interfere with the psychological analysis of other predictive tests. With respect to application in practice, psychologists could use only the HADS-A test at two time points (i.e., on day 1 and day 120), while considering the possibility of other vulnerabilities associated with pre-existing emotional instability or mental disorders.”

This manuscript aims to assess Depression and Anxiety scores in mothers of newborns carrying the TP53 p.R337H variant and to identify the possible socio-demographic associated factors. This paper is interesting, with a particular scientific soundness, well structured and clear. I have only some minor concerns.

Have you analyzed the possible impact of mothers’ age on depression or anxiety? The literature gives some important indications on the possible higher indexes in younger mothers than older ones and the possibility to answer to this research question could be assessed also in this study.

Reply

All the participants were of similar ages (>18 years) and that was the reason for not including evaluation of age difference. Paragraph in lines 250-258 was improved as, “All mothers of tested newborns lived in Paraná (South of Brazil), and no statistical difference was found in their mean ages of 26.73 ± 3.96 and 26.12 ± 6.57 (mean ± standard deviation, SD), for the control groups (non-p.R337H carriers)  C1 (without any sociodemographic variable associated with anxiety or depression) and C2 (with any sociodemographic variable associated with anxiety and/or depression), respectively; no statistical difference was found in mothers’ mean ages of 25.99 ± 6.07 and 27.91 ± 5.28 (mean ± SD) for the experimental groups (p.R337H carriers) E1 (without any sociodemographic variable associated with anxiety or depression) and E2 (with a sociodemographic variable associated with anxiety or depression), respectively.”

Another important fact that it could be inserted in the discussion could be to assess the Post-Traumatic Stress Disorder symptomatology in these mothers, a possible psychopathology in mothers subjected to such a diagnosis of their children.

Reply

Since we have not included this analysis in our study design, we introduced a sentence explaining the limitations of the study (lines 433-438).

“In addition, future studies could evaluate symptomatology of post-traumatic stress disorders in parents facing a disclosure of a genetic test positive for detrimental diseases in their newborns. For example, IES was performed in men and women after predictive test for BRCA 1/2 [38]. Similarly, we believe that future studies are needed to address the resilience and coping strategies.”

A limit is recognizable in the adopted methodology: probably a self-report measure could be subjected to social desirability and other more specific measures to assess psychological wellbeing should be adopted in the future studies.

Reply

We agree.

What about the possible resilience factors in this sense? Depression didn’t emerge as a significant variable. Perhaps resilience or coping strategies could impact on this result.

Reply

We agree. Since this is (to the best our knowledge) the first study evaluating the mothers’ psychological states after a disclosure of test results positive for cancer risk in their newborns, we believe that future studies are needed to address resilience and coping strategies.

Round 2

Reviewer 1 Report

No further comments.